# Mild-to-Moderate COVID-19 Convalescents May Present Pro-Longed Endothelium Injury

**DOI:** 10.3390/jcm11216461

**Published:** 2022-10-31

**Authors:** Paweł Kozłowski, Marcin Śmiarowski, Wiktoria Przyborska, Karolina Zemlik, Milena Małecka-Giełdowska, Aleksandra Leszczyńska, Marzena Garley, Olga Ciepiela

**Affiliations:** 1Central Laboratory, University Clinical Centre, Medical University of Warsaw, 02-097 Warsaw, Poland; 2Students Scientific Group of Laboratory Medicine, Medical University of Warsaw, 02-097 Warsaw, Poland; 3Department of Laboratory Medicine, Medical University of Warsaw, 02-097 Warsaw, Poland; 4Department of Immunology, Medical University of Bialystok, 15-269 Bialystok, Poland

**Keywords:** COVID-19, endothelium, ICAM-1, VCAM-1, E-selectin, syndecan-1

## Abstract

Background: The SARS-CoV-2 pandemic posed a great threat to public health, healthcare systems and the economy worldwide. It became clear that, in addition to COVID-19 and acute disease, the condition that develops after recovery may also negatively impact survivors’ health and quality of life. The damage inflicted by the viral infection on endothelial cells was identified quite early on as a possible mechanism underlying the so-called post-COVID syndrome. It became an urgent matter to establish whether convalescents present chronic endothelial impairment, which could result in an increased risk of cardiovascular and thrombotic complications. Methods: In this study, we measured the levels of CRP, ICAM-1, VCAM-1, E-selectin and syndecan-1 as markers of inflammation and endothelial injury in generally healthy convalescents selected from blood donors and compared these to a healthy control group. Results: We found higher concentrations of E-selectin and a lower level of syndecan-1 in convalescents in comparison to those of the control group. Conclusion: Based on our results, it can be concluded that, at least 6 months after infection, there is only slight evidence of endothelial dysfunction in COVID-19 convalescents who do not suffer from other comorbidities related to endothelial impairment.

## 1. Introduction

In May 2020, the World Health Organization (WHO) declared a pandemic of coronavirus disease 2019 (COVID-19) caused by severe acute respiratory syndrome coronavirus 2 (SARS-CoV-2) [1]. SARS-CoV-2 infection presentations may vary from asymptomatic or mild flu-like symptoms, e.g., fever, headache, muscle pain, anorexia, fatigue and (characteristic for COVID-19) anosmia, up to severe life-threatening conditions, such as acute respiratory distress syndrome (ARDS), disseminated intravascular coagulation (DIC), general hyperinflammation and multiorgan failure, which require hospitalisation and intensive treatment [2]. The virus’ high rate of spread in society, the severity of clinical symptoms and rapidly collected scientific data about the nature of this infection raised concerns about the possible consequences on survivors’ health in the future. Over time, some convalescents experienced a broad variety of persistent symptoms, e.g., cough, dyspnoea, radiological lesions, pulmonary fibrosis, chest pain, heart arrythmia, an increased risk of venous and arterial thromboembolism, chronic diarrhoea, depression, mood change, impaired cognitive function, fatigue, general muscle and joint pain, and weakness. In September 2020, the WHO established the term “long COVID” or “post-COVID” to describe health issues caused by past SARS-CoV-2 infections in all groups of COVID-19 adult survivors—those who required intensive treatment as well as asymptomatic cases [3]. It has been defined as the presence of symptoms in individuals with a history of confirmed or probable SARS-CoV-2 infection that show up usually 3 months after infection, last for at least 2 months and cannot be explained by any other reasonable cause [4,5,6,7,8,9]. Endothelial damage (“endothelitis”) caused by a broad spectrum of inflammatory mechanisms (e.g., an increase in proinflammatory cytokines, complement system and leukocyte activation) and the depletion of natural endothelial regulatory properties seem to be associated with the development of acute infection and post-COVID symptoms [10]. Comorbidities, such as obesity, diabetes and hypertension, which are associated with endothelial impairment, increase the risk of severe COVID-19 with a poor outcome [11]. The specific protein angiotensin converting enzyme 2 (ACE2), called the ACE2 receptor, was established to be an entry point for the virus to infect human cells [12]. This protein is expressed on various types of tissues, e.g., airway and alveoli epithelial cells, the endothelium, enterocytes and smooth muscle cells, as well as in the kidneys and heart [13,14]. The SARS-CoV-2 S protein (spike protein) interaction with the ACE2 receptor on the endothelial surface acts as a trigger to start the process of virus–receptor complex internalisation and cell infection. Due to massive use under these conditions, endothelial ACE2 expression is depleted, which profoundly distorts the balance in the renin–angiotensin system (RAS) [15]. It promotes vasoconstriction, inflammation and tissue remodelling with fibrosis [16]. Via the depletion of ACE2, SARS-CoV-2 can shift the renin–angiotensin system towards a proinflammatory and profibrotic state [17].

The significant decrease in nitric oxide (NO) synthesis, as well as low arachidonic acid (AA) release and prostacyclin I2 (PGI2) production, result in the inhibition of antiadhesive mechanisms. This allows platelets and leukocytes to adhere more easily, and thus promotes clot formation and neutrophil extracellular trap (NET) release [18]. An infection with SARS-CoV-2 can stimulate excessive proinflammatory cytokine release called the “cytokine storm” (e.g., IL-1α, IL-1β, IL-2, IL-6, IL-8, TNF-α and IFN-γ) [19,20]. These cytokines affect the endothelium and change its anti-coagulation properties into pro-coagulation effects [21]. Natural endothelial profibrinolytic activity is also distorted by the cytokine storm, with an increased release of the fibrinolysis inhibitor PAI-1, and a decreased synthesis of tissue plasminogen activator (t-PA) [19]. Endothelial cell activation results in the release of high-molecular-weight von Willebrand factor (vWF) multimers, while the synthesis of the ADAMTS13 enzyme, responsible for their cleavage, is decreased. This leads to the presence of huge vWF multimers in the circulation, which promotes platelet aggregation. Due to this, the activity of factor VIII is increased, thus facilitating fibrin formation [22,23]. The immune response towards coronavirus is also responsible for destroying the glycocalyx layer on the endothelial surface, which not only regulates vessel permeability and prevents platelet and leukocyte adhesion, but also activates the highly potent natural coagulation inhibitor antithrombin (AT) [24,25]. With glycocalyx impairment, all these functions are lost [26,27]. The increased expression of endothelial adhesion molecules, e.g., E-selectin (CD62E) and P-selectin (CD62P), as well as intercellular adhesion molecule 1 (ICAM-1; CD54) and vascular cell adhesion protein 1 (VCAM-1;CD106), together with the changes described above, facilitate platelet and leukocyte adhesion and fibrin formation [19]. The mechanisms briefly described above can lead to severe endothelial dysfunction, causing an increase in endothelial permeability, which greatly contributes to developing ARDS [28]. The aim of this study was to evaluate whether COVID-19 convalescents, without any additional comorbidities, develop chronic endothelial damage, which can be a trigger for further complications (e.g., thrombotic and cardiovascular incidents) in the future. For this purpose, we examined serum samples from healthy individuals and convalescents to measure the concentrations of inflammation and endothelial damage markers: C-reactive protein (CRP), VCAM-1, ICAM-1, E-selectin and syndecan-1. 

## 2. Materials and Methods

### 2.1. Control Group and Study Group

There were 294 adult participants recruited to this study among volunteer blood donors at Warsaw’s Blood Centre from August 2021 to April 2022. All of them were examined by a physician and qualified as healthy and able to donate blood in accordance with the guidelines of the Polish Minister of Health (Table 1). Additionally, they were all tested for SARS-CoV-2 infection and received a negative PCR result on the day of admission. Among all enrolled subjects, 147 of them declared previous mild to moderate SARS-CoV-2 infection at least 6 months before blood donation (the minimal period of time required between the disease and blood donation, which was confirmed by a positive PCR test result). Mild disease was defined as a lack of symptoms of lower respiratory disease (shortness of breath (dyspnoea) and abnormal chest imaging) and oxygen saturation measured by pulse oximetry (SpO_2_) ≥94%. Moderate COVID-19 was defined as symptoms of lower respiratory disease with SpO_2_ ≥94%. Another 147 individuals declared no SARS-CoV-2 infection. In the next step, all the participants were tested for antibodies against the SARS-CoV-2 N protein in their blood. Based on the anti-N SARS-CoV-2 antibody evaluation, among the 147 subjects who declared no contact with SARS-CoV-2 and no signs of respiratory tract infection, 68 had a positive result for these antibodies and were eventually classified as asymptomatic convalescents. Thus, the study group (convalescents) consisted of 215 subjects, and the control group consisted of 79 subjects. All the participants were 18–65 years old; specific data, including age and sex, as well as physical characteristics (body weight, blood pressure, etc.) were not provided. 

### 2.2. Materials and Methods

A serum sample was obtained from every participant at the day of admission to the study and stored at −70 °C until analysis. Each sample was tested and the concentrations of CRP, VCAM-1, ICAM-1, E-selectin and syndecan 1 were measured. The CRP concentrations were measured on a Dimension Exl (Siemens, Munich, Germany) analyser. For CRP measurements, a high-sensitivity particle-enhanced immunoturbidimetric assay (PETIA) was used, and the normal reference value of ≤3.0 mg/L was established. The concentrations of VCAM-1, ICAM-1, E-selectin and syndecan 1 were measured using enzyme-linked immunosorbent assays (BOSTER PicoKine^TM^ ELISA kit, catalogue numbers: VCAM-1 EK0537, ICAM-1 EK0370, E-selectin EK0501 and Syndecan-1 EK1339). For the detection and titre measurement of anti-SARS-CoV-2 N protein antibodies, expressed as COI, an electrochemiluminescence immunoassay (ECLIA) on the Cobas e801 apparatus (Roche Diagnostics, Basel, Switzerland) was used.

### 2.3. Statistical Analysis

The statistical analysis was performed with GraphPad Prism 9 software. The results of all parameters had a non-normal distribution according to the Shapiro–Wilk, Anderson–Darling, Kolmogorov–Smirnov–Lillefors and D’Agostino–Pearson tests. The Mann–Whitney U test was used for the statistical analysis of the results. The relationship between the presence of anti-SARSCoV-2 N protein antibodies and the expression of endothelial damage markers was tested with the Spearman correlation test. The results of the tested molecules and antibodies titres are expressed as the median (M), first quartile (Q1) and third (Q3) quartile [M(Q1;Q3)]. The probability value *p* < 0.05 was considered as statistically significant.

## 3. Results

### 3.1. Anti-SARS-CoV-2 N Protein Antibody Titres 

All participants were divided into two groups according to the anti-SARS-CoV-2 N protein antibody results: the control group had the negative results, and the study group had a positive result or a previous positive PCR test for SARS-CoV-2 infection. The results were defined as negative for COI <1.0 and positive for COI ≥1.0. There was a statistically significant difference in antibody titres between the two groups with *p* < 0.0001 (Figure 1). The comparison between symptomatic and asymptomatic participants within the study group showed no statistically significant difference in the median antibody titres, i.e., 23.60 (6.82; 63.60) and 15.30 (3.88; 87.98), respectively. This observation confirms that anti-SARS-CoV-2 N protein antibodies can be useful in distinguishing between convalescents and individuals who were never infected; however, it does not depend on the presence or severity of infection symptoms.

### 3.2. Comparison of CRP Concentrations

CRP, measured with a high-sensitivity method, was chosen as a possible marker of a sustained post-COVID-19 inflammatory state resulting in endothelial activation and damage. However, no statistically significant differences between the control and study groups were found for this analyte (Table 2). 

### 3.3. Comparison of Adhesion Molecules Concentrations

The median E-selectin values between the control and study groups showed a significant difference (*p* = 0.0135), with higher values in the study group. A statistically significant difference was also shown for the concentration of syndecan-1 between the analysed groups; however, a higher concentration was found in the group of non-infected subjects (*p* = 0.0082). No differences between the study and control groups were observed for VCAM-1 and ICAM-1 (Table 2, Figure 2).

## 4. Discussion

The COVID-19 pandemic has had a huge influence on people’s lives and placed new challenges on healthcare systems worldwide. Researchers all over the world have launched new investigations to understand the nature of SARS-CoV-2 virus infection, its impact on the human body and the pattern of the immune response to the virus, as well as possible long-term side effects. As the number of new COVID-19 cases increased rapidly, so did the amount of data about the role of endothelial damage in the pathogenesis of the disease. Many authors have investigated various markers related to endothelial cell activation and impairment in different groups of COVID-19 patients, divided according to the severity of the disease. Birnhuber et al. showed increased serum concentrations of E-selectin, VCAM-1, ICAM-1 and platelet endothelial cell adhesion molecule (PECAM-1; CD31) in critically ill COVID-19 patients compared to healthy controls [29]. Tong et al. tested serum concentrations of endothelial adhesion molecules (ICAM-1 and VCAM-1), proinflammatory cytokines (IL-1, IL-6, IL-18) and CRP as inflammation markers in groups of COVID-19 patients with mild and severe disease in comparison to healthy controls. They indicated that the concentration of all the aforementioned markers were positively correlated with the severity of the disease, with the highest values in the patients with a severe course of COVID-19 and lower values in the mild-disease group. The healthy control group had significantly lower values in comparison to both COVID-19 groups [30]. The same results were presented by Liu et al. [31] for the markers mentioned above and by Oliva et al. for E-selectin [32]. 

The increasing number of convalescents has created an opportunity to study the role of chronic endothelial activation and damage in developing post-COVID-19 syndrome. Haffke et al. assessed endothelial dysfunction using the reactive hyperaemia index (RHI) in patients with post-COVID-19 chronic fatigue symptoms and general weakness [33]. The RHI was found to be decreased in convalescents compared to healthy controls, indicating possible endothelium cell impairment in the first group. The same conclusion was made by Ambrosino et al. They used the measurement of brachial artery flow-mediated dilation (FMD) in COVID-19 convalescents compared to sex- and age-matched healthy controls. The authors found that lower FMD values indicated endothelial dysfunction in a group of men, whereas in women no significant differences in FMD were observed in comparison to healthy controls. The authors suggested that the protective role of sex hormones may explain these results [34]. FMD was also used by Lambadiar et al. and Oikonomou et al., who found lower values in COVID-19 patients compared to control groups [35,36]. Additional findings, such as an increased erythrocyte sedimentation rate, microparticles, homocysteine and interleukin-6 concentrations, strongly support the thesis of the destructive influence of SARS-CoV-2 infection on the endothelium. Another approach in COVID-19-related endothelial injury testing was presented by Chioh et al. [10]. In this study, the number of circulating endothelial cells (CEC) was used to assess the degree of endothelial damage. The authors found increased numbers of CEC in convalescents and a positive correlation between CEC and endothelial impairment markers, e.g., ICAM-1 and P-selectin. Since endothelial impairment triggers a subsequent reparative process, Poyatos et al. investigated the number of endothelial colony-forming cells (ECFC) as a tool to measure the intensity of this process [37]. The authors found higher numbers of ECFC in convalescents 3 months after the onset of COVID-19. Additionally, an increase in ECFC was correlated with the degree of hypoxia and high haemoglobin concentrations, as well as the male gender. These results confirm an increase in endothelial repair in COVID-19 survivors. Unfortunately, the described parameters, like many others, cannot be used in routine laboratory testing, which is why there is a constant pressure to look for widely available markers that could be easily measured to evaluate endothelial dysfunction in COVID-19 survivors. For our study, we chose CRP as a basic and routinely measured inflammation marker, which plays a role in determining the disease severity and is well established in COVID-19 patients [38,39]. Based on the fact that their endothelial expression is upregulated by the proinflammatory cytokine storm, the adhesion molecules E-selectin, ICAM-1 and VCAM-1 were selected as endothelial activation markers. Additionally, we evaluated the concentration of syndecan-1 in serum, since this proteoglycan component of the glycocalyx is shed as a result of endothelial stimulation by pro-inflammatory cytokines. The participant enrolment procedure provided the possibility of comparing the chosen endothelial marker concentrations between people who have not suffered from SARS-CoV-2 infection up to the day of enrolment and COVID-19 survivors without additional comorbidities, which could greatly influence the results. We did not observe significant differences in CRP as well as ICAM-1 and VCAM-1 concentrations between the healthy control and study groups. A possible explanation may be the time that had passed since infection. It was stated by Fogarty et al. that, during the first 10 weeks after acute SARS-CoV-2 infection, endothelial activation is common in all individuals [40]. This period of time may be extended in those who develop a severe form of the disease and suffer from other comorbidities. In our research, we tested healthy blood donors who were enrolled to the study after complete recovery. This could have provided sufficient time to diminish inflammation and re-establish the balance in endothelial metabolism, thus decreasing CRP concentrations and ICAM-1 and VCAM-1 expression and release into the bloodstream. Similar results to our observations were obtained by Tong et al. in a study performed on 345 COVID-19 survivors one year after infection. After comparing the concentrations of ICAM-1, VCAM-1, P-selectin and fractalkine in convalescents to those in healthy controls, the authors concluded that SARS-CoV-2 infection in the past does not impose an increased risk of cardiovascular events, not only for patients with a mild form of infection, but also for those who were severely ill [41].

Similar results showing positive correlations between increased CRP, ICAM-1 and VCAM-1 concentrations and disease severity were described by Liu et al. [31], Tong et al. [30] and Karampoor et al. [42]. However, the long-COVID phenomenon has raised concerns about persistent post-disease endothelial injury. This was tested in the aforementioned study by Haffke et al. in a group of patients with post-COVID-19 syndrome, who presented chronic fatigue and exertion intolerance [33]. The authors found increased concentrations of endothelin-1 in convalescents compared to healthy controls, a discovery that confirms the role of endothelial damage in post-COVID-19 syndrome. 

In contrast to the results on adhesion molecules, we found significant differences in the concentrations of the soluble form of E-selectin between the healthy control and study groups, with higher values in the latter. This finding may indicate post-COVID-19 endothelial injury; however, it was not correlated with the VCAM-1 concentration, although the expression of both of them was enhanced by pro-inflammatory cytokines. A possible explanation may be the fact that VCAM-1 can be found on a broad repertoire of cells, whereas E-selectin is more specific for endothelial cells and better reflects changes in endothelial homeostasis induced by SARS-CoV-2 infection [43]. In a study by Oliva et al., the concentration of soluble E-selectin was compared between COVID-19 patients with varying clinical disease severity: those admitted or not admitted to intensive care units, those who survived or died, and those who developed thrombosis or those who did not [32]. The authors indicated that E-selectin concentrations were significantly higher in critically ill patients who required admission to the intensive care unit, which makes E-selectin a possible predictor of disease severity. The difference between the E-selectin median concentration values in this study and our research (26.1 ng/mL vs. 1.75 ng/mL vs. 1.63 ng/mL in hospitalised patients, the study group and the control group, respectively) is noteworthy. It shows that the more severe the disease with endothelial injuries caused by the pro-inflammatory cytokine storm, the higher the E-selectin concentration. In our study, while testing generally healthy convalescents and the control group, we found low E-selectin concentrations, but these were still significantly higher in COVID-19 survivors. This finding may highlight the possible role of E-selectin as a marker of endothelial cell recovery and balance restoration. Due to scarce data about the changes in E-selectin concentrations in COVID-19 convalescents, especially in those who suffer from sustained endotheliopathy, more research is required. 

Another interesting finding is the difference in syndecan-1 concentrations between the control group and the study group with surprisingly lower values in the latter. Many studies have shown the role of syndecan-1 and its increased release from the endothelium as a marker of disease severity and poor prognosis in the acute phase of the disease. Suzuki et al. presented the case of a patient with acute COVID-19 with severe lungs involvement, whose elevated syndecan-1 concentration was correlated with their worsening clinical condition and finally decreased as the healing process progressed [44]. In the research by Zhang et al., COVID-19 patients admitted to an ICU were divided into groups of survivors and non-survivors, and the change in syndecan-1 concentration was established. The authors indicated that an increased level of syndecan-1 was a predictor of a poor prognosis since it reflected deeper endothelial injury and a more severe pro-inflammatory cytokine storm [25]. The same conclusion was made by Ogawa et al. based on their study on critically ill patients who required intensive care. These authors also concluded that the more severe clinical presentation of SARS-CoV-2 infection, the higher the syndecan-1 concentration. According to them, this is due to the inflammation process, which through many mechanisms is directly responsible for destroying glycocalyx and thus releasing syndecan-1 [45]. Additionally, Lambadiari et al. found significantly reduced glycocalyx thickness in COVID-19 as well as hypertensive patients in comparison to those of a healthy control group. The glycocalyx reduction in these groups was not correlated with the disease severity [35]. In our study the results we obtained are opposite to previously cited articles. First of all, the concentration values in the control and study groups were significantly lower compared to those in the aforementioned studies (approx. 100–1000 ng/mL versus 0.6–1.0 ng/mL). The most reasonable explanation is the type of study group. We focused not on critically ill patients admitted to intensive care units, but on generally healthy convalescents in comparison to the control group. Our participants were free from other comorbidities which could be related to chronic endothelium activation and injury. This fact can explain the much lower syndecan-1 level that we obtained in our study. However, the difference in the concentration of syndecan-1, with a surprisingly lower level in convalescents, needs to be explained. The role of the heparan sulphate proteoglycans group in different cells’ biology, including in wound healing, angiogenesis and malignancy, has been studied for many years. A syndecan-1 molecule expression on endothelial cells has been linked especially to regenerative and angiogenesis processes. In 1996, Kainulainen et al. published their study, in which they were testing the influence of different pro-inflammatory cytokines on syndecan-1 endothelial expression [46]. The authors concluded that among many tested cytokines, only TNF-α was able to inhibit syndecan-1 gene expression, thus decreasing this protein quantity on the endothelial cells’ surface. This influence was observed especially during regeneration. As syndecan-1 is believed to take part in the healing process, its downregulation might have a negative impact on restoring homeostasis in the endothelium. However, Javadi et al. indicated that syndecan-1 overexpression can also result in the inhibition of endothelial cell proliferation and the restoration of physiologic balance in the endothelium [47]. Despite some sparse reports that may support such a hypothesis, we still believe that the observed lower syndecan-1 concentrations in convalescents need to be explained and the possible mechanisms of its downregulation need to be established. 

Knowledge about different pro- and anti-inflammatory cytokines changes and the effects they have on cells seems to be crucial in understanding the mechanism underlying post-COVID-19 endotheliopathy with possible complications. One limitation of our study was the inclusion of patients who suffered from COVID-19 only; it has to be mentioned that other acute systemic infectious conditions may induce endothelial injury as well. However, our aim was to assess the possible effect of SARS-CoV-2 infection on endothelial function in generally healthy subjects. Nonetheless, it might be interesting to compare how different types of systemic inflammation may affect endothelial function. Another limitation of our study is the lack of data about participants’ sex and BMI; however, we believe that both the study and control groups can be regarded as homogenous due to the very strict criteria for blood donation.

## 5. Conclusions

Based on our results, it can be concluded that, at least 6 months after infection, there is only slight evidence of endothelial dysfunction in COVID-19 convalescents who do not suffer from other comorbidities related to endothelial impairment. The increased E-selectin concentrations in convalescents, which may be associated with prolonged endothelial injury, require further investigation and confirmation.

## Figures and Tables

**Figure 1 jcm-11-06461-f001:**
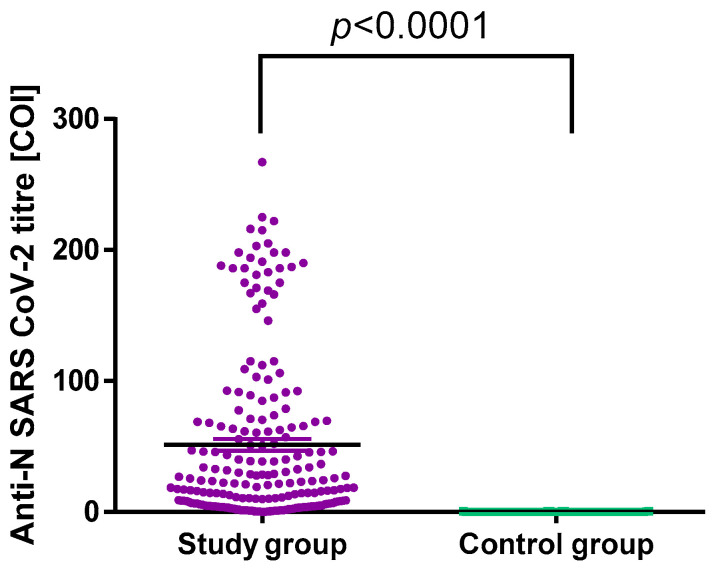
Anti-SARS-CoV-2 N protein antibodies titres in the control and study groups.

**Figure 2 jcm-11-06461-f002:**
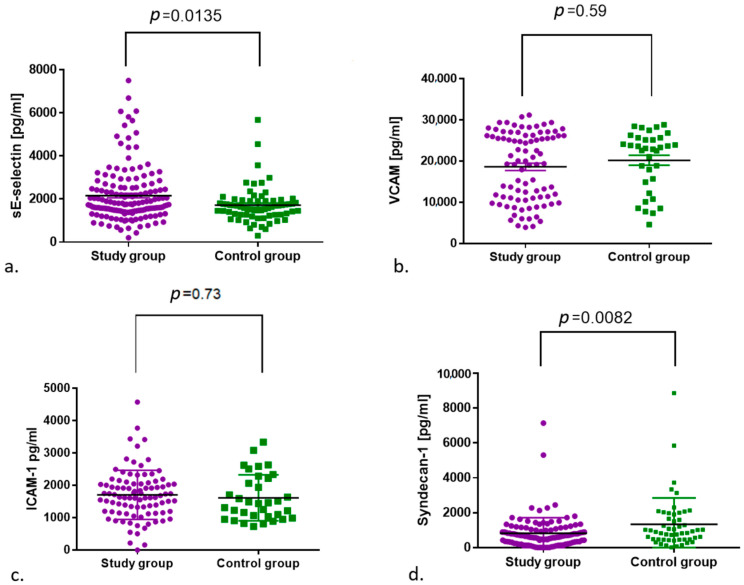
Comparison of (**a**) E-selectin and (**b**) VCAM-1, (**c**) ICAM-1 and (**d**) syndecan-1 concentrations between the control and study groups.

**Table 1 jcm-11-06461-t001:** Exclusion criteria for the study.

increased bleeding risk.
acute or chronic disease of: the cardiovascular system, central nervous system, gastrointestinal tract, airways and lungs, genitourinary system, immune system, endocrine system, skin
acute or chronic connective tissue disease
diabetes
any type of cancer, now or in the past
infectious diseases: positive PCR for SARS-CoV-2 at the day of admission HBV, HCV, any hepatitis of undetermined cause HIV-1/2, HTLV any parasitosis patients with any risk of transmissible spongiform encephalopathies (TSE) syphilis at any time in life history of drug or alcohol abuse history of risky sexual behaviour sex workers any sexually transmitted diseases (STD) in the past allotransplant recipients increased body temperature in the last several weeks any vaccination in the last 4 weeks any medical or non-medical procedure with risk of infection (e.g., surgery, tattoo, etc.) in the last 6 months travelling in the last 6 months to countries in which there is an increased risk of infectious diseases

**Table 2 jcm-11-06461-t002:** Median concentrations of soluble adhesion molecules and CRP in the study and control groups. There were significant differences in E-selectin and syndecan-1 between the two groups.

	E-Selectin [pg/mL]	ICAM-1 [pg/mL]	VCAM-1 [pg/mL]	Syndecan-1 [pg/mL]	CRP [mg/L]
Control group	1633 (1272; 1918)	1465 (1034; 2065)	40,341 (23,578; 54,718)	934 (466; 1944)	2.4 (2.0; 3.2)
Study group	1754 (1422; 2520)	1738 (1338; 2157)	41,959 (24,738; 50,490)	692 (342; 1138)	2.5 (1.9; 3.0)
	*p* = 0.0135	*p* = 0.73	*p* = 0.59	*p* = 0.0082	*p* = 0.36

## Data Availability

The data presented in this study are available on request from the corresponding author.

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
