# Peer review of "Mild-to-Moderate COVID-19 Convalescents May Present Pro-Longed Endothelium Injury"

_jcm, 2022, doi:10.3390/jcm11216461_

Round 1

Reviewer 1 Report

Thank you for the opportunity to review this original article on such a timely topic. It should be noted that other studies have touched upon endothelial injury in convalescent COVID-19, so the originality of this paper cannot be considered high. In this version, the paper is not well-organized and needs to be heavily revised.

Major comments:

1. Introduction looks like a review article, very extensive together with figures. The authors should significantly shorten this part, maybe by moving some information to the discussion section and removing the figures.

2. The study design needs additional clarity. Authors state that among participants that declared no contact with SARS-CoV-2, the presence of positive antibody testing classified them as asymptomatic convalescents. An important question that arises from this statement is whether the subjects were vaccinated. If so, this could be the result of vaccination.

3. Clinical characteristics of the patients are lacking (age, sex, BMI, blood pressure, etc), and thus we cannot assume if these factors may account for the differences between the study and the control group. Moreover, the time period of the infection relative to the date of blood donation is not specified.

4. Another introduction into adhesion molecules is not necessary in the results section. Please delete it or move it to the discussion.

5. The discussion section in its current format is not comprehensive. Try to divide it into smaller sections. Moreover, important studies are missing (10.1016/j.vph.2022.106975,10.2174/0929867328666211026124033,10.1002/ejhf.2326, etc).

6. English language should be assessed by a native speaker.

Minor comments:

1. Remove tables 2 and 3; they are redundant.

2. Consider merging tables 4 and 5.

3. NTproBNP and hsCRP are not markers of endothelial injury. Please revise.

4. Add p-values also for non-significant associations, in tables and figures.

Reviewer 2 Report

I carefully read the original research article entitled “Mild-to moderate COVID-19 convalescents may present pro-2 longed endothelium injury” from KozÅ‚owski et al. Both the topic and the protocol study investigated are of relevance and interest. I did not encounter any major concerns. One suggestion I would make to the AA. is to revise the article, avoiding the repetition of concepts already expressed. The inclusion of bibliographic references, such as on the level of damage to the immune system, is also strongly recommended. For the rest, I can only congratulate the AA. for the interesting study conducted.

Author Response

Please see the attachemnt.

Reviewer 3 Report

The paper by Kozłowski et al. deals with a very interesting topic: the Authors investigated whether persistent underlying endothelial injury may be detected in COVID-19 convalescents after recovery. In order to address this issue, the Authors chose to measure the values of CRP, BNP, ICAM-1, VCAM-1, E-selectin and syndecan-1 as markers of inflammation and endothelial injury, highlighting higher concentrations of E-selectin and lower levels of syndecan-1 in convalescent subjects in comparison to controls.

Major comments

The inclusion criteria should be discussed more into details. The Authors mention “mild-to moderate SARS-CoV-2 infection (at least 6 month before blood donation)”: a detailed description of how “mild-to-moderate infection” was identified and of the reasons why this specific timing was chosen is warranted.

Was there really a solid background to hypothesize alteration of these biomarkers after such a time span ? Is it possibile to provide the specific median time after infection for the population under investigation (or at least the time range) ? Even though the cohort is composed of otherwise healthy sujects, is it possible to be sure that no other confounding factors affected such measurements after such a time span ?

The conclusions seem to be confusing, almost merely speculative and not supported by the results presented in the current manuscript. In my opinion, what is reported in the conclusions might be moved to the discussion section, perhaps softening the sentences and underlining that “the risk of developing cardiovascular disease” in the different subgroups of subjects (symptomatic and asymptomatic convalescents as compared to the control group) should be investigated by further research.

A thorough revision of the different paragraphs is required, since introduction, materials and methods, results, discussion overlap and this is not acceptable. Some examples are the following:

- page 4, lines 94-99: In the Introduction section the Authors anticipate some of the results and also report a comment which should actually be part of the Discussion: “We found no significant difference between healthy control and convalescents in the concentration of tested markers except the E-selectin and syndecan-1, the molecules which are well recognized as the marker of endothelium activation and impairement. This finding indicate that there is a possibility of sustained process of endothelium activation after SARS-CoV-2 infection”. Please, delete these sentences from the Introduction.

-page 9, lines 172-176: “An infection with SARS-CoV-2 virus causes the release of proinflammatory cytokines which interact with endothelial cells increasing an expression of adhesion molecules. The endothelial activation may sustain long after the infection thus playing an important part in long-COVID syndrome. Due to that the adhesion molecules which are initially expressed on the endothelium and next are released into the bloodstream, they may be used as markers of endothelium activation and damage in COVID-19 convalescents.” This actually belongs to the introduction and background and should not be repeated in the Results section.

-page 9, lines 176-177: “The concentrations of four adhesion molecules were measured in this study: sE-selectin, ICAM-1, VCAM-1 and syndecan-1.” This has been already explained in the Methods section.

-page 12, lines 237-241: “We obtained serum samples from healthy donors who visited blood bank for honorary blood donation. At the day of admission to the blood bank and serum sample collection all of the participants were examined by a physician and evaluated as being healthy and able to become blood donors. In the next step we divided participants into two groups: healthy control and convalescents study group based on the result of anti-SARS-CoV-2 N protein antibodies with negative result and positive result respectively.” This is a redundancy since this has already been explained in the methods section. In case the Authors want to highlight this method as a strength of the study, this should be explained otherwise (and hopefully more shortly).

-Page 12, lines 257-259: “It has been proven in many research that SARS-CoV-2 infection leads to the endothelium activation and damage. In the study by Qin et al. a degree of CRP and BNP concentration increase in acute phase of COVID-19 was correlated with the disease severity.” This actually belongs to the introduction. 

The reasons why the Authors chose to measure BNP as a marker of endothelial damage should be further justified. Moreover, the reference mentioned (number 37) actually refers to critically ill COVID-19 patients and analyzes several prognostic markers, with no specific correlation with endothelial injury.

According with the introduction and in consideration of the importance of the prothrombotic state induced by SARS-CoV-2 infection (with important clinical implications), it would be more straightforward to test  markers related with the so-called immunothrombosis.

In my opinion a weakness of the study is the lack of comparison with patients recovered from other kinds of acute systemic infectious conditions (e.g. ARDS of other etiology). Is the endothelial injury described specific to SARS-CoV-2 infection ? I think that, even though the experimental design did not imply comparisons with other inflammatory conditions, a quick comment on this point is deserved.

The speculation on the reasons why syndecan-1 showed unexpectedly lower values in convalescents as compared to controls is slightly obscure. The Authors seem to underline the involvement of syndecan-1 in healing processes in order to hypothesize a sort of consumption in convalescents due to enhanced regenerative processes. However, the explanation sounds quite difficult to understand and not clearly supported by the literature. References should be added and some sentences which seem not to be consistent with the above-mentioned hypothesis should be rephrased (e.g. page 14, lines 311-313 “The phenomenon of lower syndecan-1 and higher E-selectin concentration in convalescents without additional comorbidities may be explained as a slight but chronic post-COVID-19 inflammation affecting endothelial cells”).

Even though this is not the precise focus of the paper, I think that a comment is deserved on alteration of endothelial permeability in COVID-19, given its importance in the pathogenesis of the condition.

Page 6: among the exclusion criteria the Authors do not mention “history of alcohol abuse” or “recent travels abroad”, which are actually also commonly used exclusion criteria in many countries. Were these criteria included ?

The acronym STD should be explained as sexually transmitted disease.

Page 6, lines 115-117: “Thus, the study group (convalescents) consisted of 215 subjects, and control group of 79 subjects. All the participants were at the age of 18-65 years old, all their specific data including age and sex were anonymous. Groups characteristics in presented in Table 2.” Here the Authors mention that Table 2 reports some data, which are actually not presented in the Table (on page 7). Please, complete Table 2 adding the characteristics of the population (gender, age + BMI, vital parameters etc).  Moreover, the term “asymptomatic” and “symptomatic” should be better explained: which symptoms were mostly represented and in which proportion or combination?

Moreover, a weakness of Table 2 is the lack of the p values regarding both the comparison between the control group and study group and the comparison between the two subgroups of the study group.

Page 7, line 133: “The normal reference ranges for these molecules were no established.” This is a point which may raise many doubts. In my opinion, even though these are markers which are mostly used for reasearch rather than in the clinical practice, some references should be provided on their alterations, at least in similar clinical scenarios. Otherwise, the reader, especially if not expert in the field, may find it difficult to give a specific value to increases of these markers and may not understand the possibile clinical relevance of specific increases. In our everyday clinical practice we are used to attribute value to “delta” of some laboratory parameters not only from the statistical point of view, but in the context of correlated alterations of the clinical picture. Therefore, some tools should be provided to understand the results presented.

Page 8, line 160: “Comparison of Cardiovascular Damage Markers Concentrations”: I think that this title is misleading because while BNP is certainly a cardiovascular marker, CRP can only indirectly be considered as marker of cardiovascular injury. Perhaps, the Authors can consider to add some references on the correlation of high sensitivity CRP with cardipvascular damage in the introduction.

Page 8, lines 152-154: “This observation confirms that anti-SARS-CoV-2 N protein antibodies can be useful in distinguishing between convalescents and individuals who were never infected, however it does not depend on the presence or severity of infection symptoms.” Please, rephrase (especially the last part of the sentence whose meaning is clear but the wording could be substantially improved)

In the Discussion the Authors mention some studies on flow-mediated dilation in COVID-19 convalescents, but the interpretation they provide of the results of these studies is misleading.

Page 11, lines 214-216:” The authors found higher FMD values to indicate endothelial dysfunction in a group of men, whereas in women no significant differences in FMD were observed in comparison to healthy control.” The Authors actually found LOWER FMD values in men. Otherwise, the results would have been unexpected and would have not suggested endothelial dysfunction.

I suggest to mention reference 32 after explaining the results, otherwise it may be suggested that Ambrosino et al. also used the RHI, while the following sentence (after ref 32 in the current version of the manuscript) explains that they actually used FMD.

Page 11, lines 217-219: “Jud et al. also used FMD to assess endothelial damage in convalescents [33]. They stated, in contrast to Ambrosino et al., that high FMD values are indicative for endothelial cells impairment for both groups: 218 men and women.”

The paper mentioned actually does not provide the results mentioned. Jud et al. actually found no difference for FMD among the three groups considered (COVID-19, ASCVD and controls). They only comment on the fact that post-COVID-19 patients had similar values of FMD compared to patients with ASCVD, “which may be attributed to other subject-related factors influencing vascular reactivity, like smoking, physical activity, mental stress, alcohol intake or hormonal changes during physiological menstrual cycle”. They do not discuss esplicitly gender-specific differences.

I think that a paragraph on the limitations of the study is certainly needed.

Figure 2 requires a detailed legend. Moreover, even though “glycocalyx damage” is indicated in the “zoom circle”, in the main picture there is nothing suggesting the presence of the glycocalyx, which should be preferentially added. I would suggest to indicate “syndecan-1”, since this is the biomarker they chose to measure.

Minor comments

Thorough English revision is required.

Surprisingly, the term “epithelium” seems to be often used instead of “endothelium”, even in the abstract.

Moreover, “endothelium” is often used instead of “endothelial” (e.g. “endothelial injury” is more comonly used than “endothelium injury”).

The word “glycocalyx” is repeatedely misspelled, with “i” instead of “y”

The term “viral” is often erroneously written as “virial”.

Subjects and verbs must agree with one another in number (singular or plural). Please, check for consistency because there are several examples of lack of concordance throughout the manuscript. Moreover, singular and plurals have to be checked also independetly of the verbs because the singular form is often used when the plural would be better.

Please, check also articles and prepositions.

Acronyms should be always explained the first time they are used, while, for instance, on page 7 the meaning of some acronyms is explained after they have already been used in the previous pages.

For the sake of homogeneity, in the manuscript figures should be indicated either as Fig. (e.g. Fig. 1 and 2) or as “Figure” (e.g. Figure 3)

The term “that” is often erroneously preceeded or followed by a comma.

Page 4, lines 84-87: “The immune response towards coronavirus is also responsible for destroying the glycocalyx layer on endothelium surface, which not only regulates the vessels permeability and prevents platelets and leukocytes adhesion, but also activates very potent natural coagulation inhibitor-antithrombin (AT).” I suggest to add a reference for this sentence.

Page 7, line 127: here the Authors mention “sE-selectin” while the same marker is often mentioned simply as “E-selectin”. Please, correct for the sake of homogeneity.

Pge 11, line 201: “Ming et al.” vs page 12, line 252 “Tong et al”: please, check name and surname and refer to the Author always mentioning the surname.

Page 11, line 211: “The RH index was found to be….” The acronym RHI should be used, as mentioned in the previous sentence.

References should be checked thoroughly. Some references report the name + surname of the first Author (e.g. reference 23, 31 and 33), while in others the first name is presented instead of the surname (see comment above)

Round 2

Reviewer 1 Report

The authors have satisfactorily addressed all of my comments.

Author Response

Dear Reviewer,

thank you for all your comments.

Kind regards,

Olga Ciepiela and co-authors

Reviewer 3 Report

In the revised manuscript the authors have satisfactorily addressed most of my comments and  concerns raised on their original submission. This revision has significantly improved the paper, which is suitable for publication in the current version.

Author Response

(The authors gave the same response as above.)
